# Conversational, Longitudinal, Ecological Assessment (CLEA): Exploring a new AI-driven method for qualitative data collection in a behavioural health context

Samuel Downes[1]*, Thomas Krys[2], Kenton O'Hara[3], Max Western[4], Lauren Thompson[5], Amberly Brigden[6]

**1** School of Electrical, Electronic and Mechanical Engineering, University of Bristol, Bristol, United Kingdom, **2** Bristol Medical School, University of Bristol, Bristol, United Kingdom, **3** School of Computer Science, University of Bristol, Bristol, United Kingdom, **4** Department for Health, University of Bath, Bath, United Kingdom, **5** School of Engineering Mathematics and Technology, University of Bristol, Bristol, United Kingdom, **6** School of Engineering Mathematics and Technology, University of Bristol, Bristol, United Kingdom

* s.downes.2021@bristol.ac.uk

## Abstract

In this paper, we present conversational longitudinal ecological assessment (CLEA), a novel conversational AI–enabled method for collecting ecologically valid, temporally sensitive qualitative health data via mobile instant messaging. We report findings from an exploratory deployment of an instantiation of CLEA within a 12-week community-based weight management programme, delivered by a charity partner in an area of deprivation. Using WhatsApp, we deployed our CLEA chat-agent to conduct twice-weekly conversational data collection sessions with participants, to elicit data about their experience of the programme and associated behaviour change. This was followed by in-person semi-structured interviews (N = 9) to examine user experiences and perceptions of interacting with the chat-agent. Participants reported that WhatsApp's familiarity supported accessibility and sustained engagement, while the conversational format encouraged reflection directed towards the research focus. Responding to chat-agent prompts required cognitive effort, leading some participants to defer engagement until they had adequate time and mental space; however, this reflective demand was largely experienced as beneficial within the programme context. The AI's quasi-human interactional qualities fostered a sense of support while reducing social judgement, enabling more candid disclosure. Together, these findings suggest initial feasibility and acceptability of this CLEA implementation within a community-based programme in an area of deprivation. Further, while the responses in single messages were often brief, useful, relevant, and meaningful insights appeared to develop over the course of conversational sessions. The study highlights both the opportunities and trade-offs of conversational AI for qualitative data collection, including design implications for health researchers looking to

**Data availability statement:** Considering the highly sensitive personal qualitative data in our study, in line with participant data protection due to risk of re-identification, our research data is not available on a publicly held data repository. However, anonymised data is available on request, held in the university of Bristol's research data storage facility https://www.bristol.ac.uk/acrc/research-data-storage-facility/ Requests for partial or completely anonymised data may be made through contacting rdsf-help@bristol.ac.uk.

**Funding:** Funding from an EPSRC studentship (Grant code: EP/W524414/1) allocated to lead researcher SD was utilised to complete this work. The funders had no role in study design, data collection and analysis, decision to publish, or preparation of the manuscript.

**Competing interests:** The authors have declared that no competing interests exist.

implement or extend the method. Finally, we position CLEA in relation to other longitudinal methods of qualitative health data elicitation.

## Author summary

Developing effective interventions for health behaviours such as healthy eating and physical activity requires methods that can capture the complex, individual factors shaping people's everyday experiences, including stress and motivation. Because such factors often fluctuate over time, longitudinal approaches are needed to understand how experiences and behaviours unfold in real-world contexts. For such methods to be effective, they must also be acceptable, engaging, and accessible; particularly for underserved or disadvantaged populations known to experience lower health and digital literacy. In this study, we introduce conversational longitudinal ecological assessment (CLEA), a digital health method that uses conversational AI technology to collect ecologically valid qualitative data over time through an accessible instant-messaging platform. We demonstrate the feasibility, acceptability, and utility of CLEA through a real-world deployment investigating the experiences of individuals accessing a community-based weight management programme, being delivered in an area of deprivation. To support other health researchers, we position CLEA in relation to existing longitudinal qualitative methods and highlight the key design considerations that shape engagement, data quality, and participant experience.

## 1. Introduction

Qualitative research is central to understanding complex health events, phenomena, and systems. Detailed insights into individual or group experiences can illuminate not just 'what' outcomes occur but also offer deeper understanding of 'how' and 'why' they occur [1]. Such experiences are subject to fluctuations over time, being affected by a complex interplay of personal, social and environmental factors that exert dynamic effects over a given period [2]. Capturing the full range and depth of the attitudes, motivations, moods, emotions, behaviours, and socioenvironmental factors that shape people's experiences over time is a significant methodological challenge [3]. Collecting data that is simultaneously insightful, relevant, ecologically valid and representative of an individuals' authentic fluctuating experiences is similarly difficult [4]. This methodological balancing act must also account for the needs of respective participants. Researchers therefore face trade-offs in method selection, that varies according to research context, phenomenon of interest and resource availability.

### Methods for temporally sensitive, qualitative data collection

A common method for capturing temporal change is the interview - an in-depth conversation at a single time point that can produce rich, authentic narratives [5]. While

skilled interviewers can build trust and elicit high-quality data through adaptive, dynamic techniques, data is reliant on participants' recollections, afflicted by imperfect recall or social desirability [5]. Repeating interviews with the same participant over time can mitigate recall issues; though at significant resource cost [6]. Additional methods can supplement interviews, such as collecting data from wearables, providing ecologically valid, longitudinal data to prompt recall [7]. Doing so, however, requires considerable setup costs, onboarding, and potential acceptability concerns around intensive tracking [8]. Beyond interviews, distinct intensive longitudinal methods exist for temporally sensitive research [9]. End-of-day diary studies (handwritten or digital) bound recall to a daily interval, capturing participants' narrative summaries and interpretive appraisals of experiences unfolding across the day [9–11]. Considering data provision is self-led, it can vary in relevance and clarity, increasing analytic demands for researchers [12].

Another longitudinal research method, ecological momentary assessment (EMA), involves prompting participants more frequently about momentary experiences, to elucidate fast, transient, and context-sensitive phenomena [4]. To enable this, many designs rely on constrained response formats (e.g., short-answer items or quantitative scales); with the trade-off of limiting richer disclosure) [13,14]. EMA also typically requires considerable onboarding, and close compliance management due to the burden imposed on participants by the prompt frequency [4]. In studies using EMA for behavioural health research, compliance has been demonstrated to decline linearly over study duration [14]. In addition to extending the acceptability and engagement of longitudinal methods, methodological value could lie in methods that combine the interpretative richness of end-of-day diaries, with the directive, topical focus characteristic of EMA designs [15]. Such methods could be appropriate for research areas such as behavioural health research, where individually complex determinant factors fluctuate over time and are strongly shaped by personal context; and where EMA designs may miss important contextual nuance or interpretation [16–18].

In summary, qualitative interviews enable adaptive, in-depth data collection but are resource-intensive, often infrequent and removed from the moment of experience. Intensive longitudinal research (ILR) methods allow frequent, in-context data capture closer to events, but are static, varying from constrained (EMA) to unstructured (diaries). Integrating conversational AI into an IRL approach, offers the potential to integrate some of the strengths of interviews - through enabling flexible adaptive interactions with enough structure to stay on topic - while enabling repeated, in-the-moment data collection.

## Conversational agents (CAs) for qualitative data collection

There has been growing interest in recent years in the application of artificial intelligence (AI) to qualitative research. While much prior work has focused on using AI to support qualitative data analysis [19–21], comparatively little attention has been paid to how it might be used in data collection. This is surprising considering the potential of conversational AI to collect data in a directive, engaging manner that makes space for interpretive richness. Initial studies have demonstrated the feasibility of AI-powered conversational agents (CAs) for conducting qualitative interviews or inquiry, most often in single encounter or non-naturalistic laboratory settings [22–25]. To date, however, there has been limited investigation into the deployment of CAs for longitudinal, in-the-wild qualitative data collection, or their methodological implications relative to EMA, diaries, or interviews. Specifically, little is known about how the acceptability, feasibility, or utility of AI-mediated qualitative agents might vary over time or be influenced by design choices. A brief overview is provided below of CA technologies and existing studies exploring their potential in qualitative data collection.

## Evolution of conversational agents (CAs)

CA technologies have co-evolved with developments in AI over the last century. Early systems used hand-crafted rules to emulate a sense of conversation, albeit highly constrained [26]. An example of such a chatbot deployed for qualitative data collection is Tallyn et al.'s "Ethnobot" [27], used to gather experiential, momentary qualitative data during a public event. While users of this early system frequently anthropomorphised it, many were disappointed by its repetitive

behaviour or frustrated when it did not meet their conversational expectations. This is aligned to a phenomenon described as the "gulf in user expectations" [28]. More advanced CA technology combines machine-learning capabilities alongside older rules-based technology, in hybrid systems [26]. Xiao et al. [29] compared performance of a hybrid CA to traditional fixed-question survey, with participants (video-gamers) providing feedback on video-game trailers. They gathered more informative data from the CA enabled tool, and participants expressed greater positive sentiment towards the CA. While video-gamers are more likely technophiles, this work nevertheless highlights how hybrid CA systems can be effective for qualitative data collection within pre-specified topics, potentially outperforming traditional fixed-question surveys.

More recently, with the emergence of large language models (LLMs), CA systems are now capable of more dynamic conversation, sustaining coherent, dialogue with greater generalisability to different topics and use-contexts [29]. Several studies have investigated their use for qualitative data collection. A large-scale study (n = 399) comparing CA technologies for qualitative data elicitation found that despite simpler development requirements, a lightweight LLM-based CA elicited similarly informative data and was more favourable to users than a sophisticated hybrid CA [30]. Geiecke and Jaraval [23] conducted a rigorous examination of a lightweight LLM-based CA by recruiting sociologists to score transcripts of AI-led interviews, reporting data obtained as on-par to an "average human expert" interviewer. They also found participants expressed preference for AI-led interviews over human interviews, as the AI was non-judgemental.

Advances in the conversational ability of CAs have demonstrated the possibility of conducting engaging, single-session conversational data collection remotely at scale [23–25,29]. The work of Xiao [29], Cuevas [31] and Geiecke and Jaraval [23] highlights how these newer forms of CA technology may narrow the gulf in user expectations that limit richer disclosure in earlier generation CAs, like Tallyn's 'Ethnobot'. The previously detailed works demonstrate that the capability of CAs to engage in interactive and dynamic conversation led to more informative, detailed responses from participants relative to static, traditional survey approaches. Whether this utility extends to longitudinal methods in health contexts is yet to be established.

To address this research gap, we present an exploratory deployment of a new method for conversational, longitudinal, ecological assessment (CLEA), that utilises conversational AI and mobile instant messaging. We position CLEA as a technological adaptation to ILR methods that, through specific design choices, can be adapted to conduct longitudinal research with varying qualities of EMA and/or diary method depending on the requirements of the respective study context. In this work, we explore the use of CLEA for qualitative data collection in a behavioural health context. The presented instantiation of CLEA employs LLM-based conversational AI and WhatsApp to gather twice-weekly qualitative data over a 12-week weight management health intervention programme (henceforth referred to as 'the programme') in an area of deprivation. Our research aims in this work were to establish A) feasibility of the tool B) acceptability of the tool to users, and C) the utility of the data captured over this period. We adopt a pragmatic approach for guiding methodological decisions that embraces the co-constructive nature of qualitative data, and the interpretative manner of qualitative analysis. Through our findings, we evidence the potential value of this technology-mediated method as well as its constraints. Drawing on this analysis, we contribute early insights into the design of next-generation tools for conversational longitudinal ecological assessment research.

## 2. Methods

### 2.1 Study design

This is a mixed-methods study, with quantitative and qualitative components undertaken in parallel. Different methods were chosen to address the different research aims: quantitative analysis of the chat logs to explore longitudinal engagement (aim A); content analysis of chat logs to explore clarity, relevance, depth and utility (aims A and C); and analysis of qualitative interviews to explore users' experiences and perceptions around acceptability, usability and utility (aims B and C)

## 2.2 researcher positionality

The research team comprised the first author, who was closely involved in the development, deployment, and evaluation of the CLEA system; a clinician who was more removed from system development, but had an interest in applied digital health research; and three senior researchers with expertise in digital health and qualitative methods. The analysis of interview data and the rating of chat logs involve an element of subjectivity, the team's diverse composition enabled multiple perspectives and ongoing sense-checking throughout these processes. This helped ensure that the analysis remained grounded in the data and was not unduly influenced by any single researcher's perspective.

Regarding the team's epistemological stance on AI-mediated qualitative research, we adopted a pragmatic stance that embraced the constructive nature of human and AI-mediated qualitative data generation, and the interpretive elements of qualitative analysis. In particular, the team recognised the need for reflexivity in CLEA's role in data-production, allowing for consideration of how researcher design choices might influence data produced by the method.

## 2.3 Setting and context of CLEA deployment

This study was conducted in collaboration with the Robins Foundation - a local charity associated with Bristol City Football Club, providing health and wellbeing support to people from areas of deprivation in Bristol, UK. The Foundation deliver a 12-week behavioural health programme, targeting weight loss, in an area of deprivation. Within this programme, we deployed an early version of CLEA, to gather data regarding (1) participants' immediate feelings after attending a health session to understand their experience; and (2) the barriers/facilitators participants experience in applying session knowledge and skills in their daily life. Features of this instantiation of CLEA were tailored according to collaborator advice, providing alignment to the needs of programme participants. Through this, we aimed to shed light on the acceptability (in terms of participant experiences, perceptions and attitudes), feasibility (terms of instrument fidelity and data completeness), and utility (how useful the data is over time) of conversational, longitudinal, ecological assessment in a behavioural health context.

## 2.4 Participants and recruitment

Individuals were eligible to participate if they were: (1) at least 18 years old, (2) English-speaking, (3) had access to a smartphone with WhatsApp, (4) were in stable housing, (5) no critical health conditions and (6) engaged in the programme. Baseline socio-demographic information was collected, and participants received onboarding instructions for the CLEA system. Sampling was opportunistic via the programme. Most participants (n = 5) received study information in week 1, were invited to provide informed consent from week 2, and were onboarded from week 3. Additional participants were onboarded in weeks 4 (n = 3) and 5 (n = 1). This staggered approach was chosen to increase access to participation for those who missed early programme sessions or required more time to consider participation. However, it resulted in a shorter period of technology use for some participants.

## 2.5 CLEA system

Technology choices were guided by principles of performance and safety, with consultation from charity partners, advocating for user needs. Considering performance, model selection was constrained by the technical landscape at the time of development (late 2023). Within this context, GPT-4 Turbo demonstrated reliability in consistently attending to information across longer prompts, follow system instructions reliably, maintain conversational coherence, and remain aligned to the intended research scope [32,33]. GPT-4-Turbo was prioritised for prototype testing with charity partners in these respects, considering its preferable performance in terms of fidelity, which has implications for safety – for example, fidelity to adhering to guardrails like do not give medical advice. A prototype was set up in a test environment and connected to WhatsApp installed on university-owned mobile phones. Charity staff attended an in-person testing session, during which

they interacted with the prototype live and provided feedback on its perceived suitability. In particular, they viewed the system's positive and supportive tone as appropriate for the participant group. Other feedback such as number of questions per turn, and probing constraints were implemented in the prompt instructions.

WhatsApp was identified as a suitable interface because partners already used WhatsApp Groups to communicate with participants during programme delivery, suggesting that it would be familiar and accessible within this context.WhatsApp is easy-to-use, cheap, and familiar to the majority of iPhone and Android smartphone devices. Being available over WiFi networks, WhatsApp enabled cost effective use for those without cellular data plan. Furthermore, onboarding was straightforward, identifying users via their telephone number rather than requiring a password-protected account. Partners endorsed this as an inclusive choice for user interface; they themselves use WhatsApp for group communications, finding it to be highly accessible to those with lower digital literacy. The WhatsApp interface can be seen in Fig 1.

The WhatsApp front-end enables participants to send and receive messages to our chat-agent, via our custom web application, which forwarded messages to and from our selected LLM, OpenAI's GPT4-Turbo. Chat-logs from these

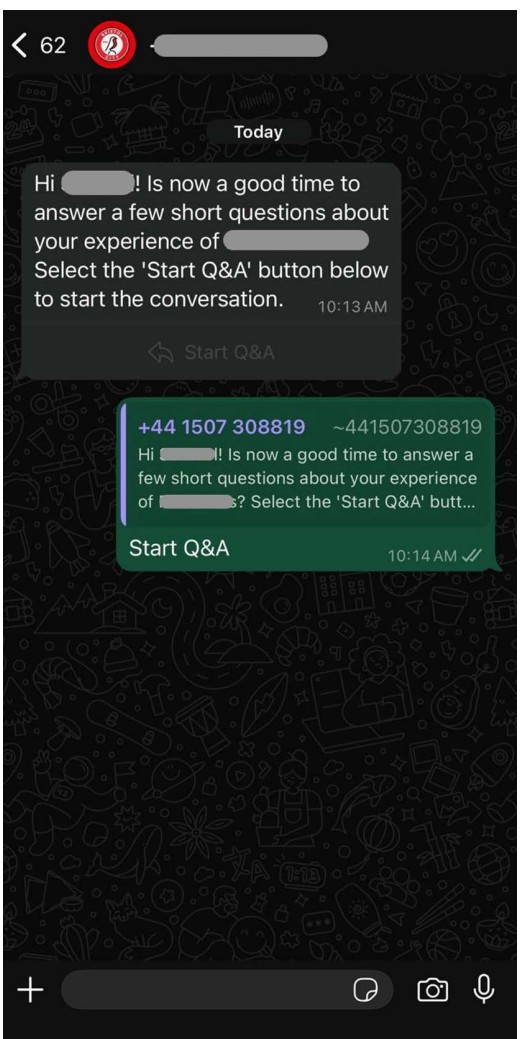

**Fig 1. Sample visual of user interface in responding to a single data collection prompt by the CLEA system in WhatsApp.**

messaging sessions would be periodically downloaded via an encrypted connection, to a password protected Excel spreadsheet hosted on the University's encrypted Microsoft SharePoint site. In terms of piloting the acceptability of the AI, partners appreciated its generally supportive tone, feeling it would be appropriate for programme participants.

The choice of LLM 'under the hood' is fundamental. Beyond demonstrating suitable fidelity, the GPT4-Turbo model tends towards a positive, affiliative tone and leans towards 'safe' interactional practices - avoiding overly sensitive or emotional topics [34,35]. This tendency towards 'safe' interactions was desirable in this instance, considering typical in-person 'distress protocols' such as reading non-verbal cues are not applicable here [36].

Our application contained specific prompt-based instructions to guide the AI in managing conversation flows with participants. Feedback from our charity partners was implemented within these prompt instructions following testing. Fig 2a–2c detail the core operating instructions and research targets for the question-answer sessions. The chat-agent was instructed to act as a reflective interviewer, positioning it to elicit personal reflections. The prompt instructions prioritised understanding the participants' feelings, experiences, and behaviours. This, in effect, guided the content and frames of dialogue. Based on partner advice to manage participant burden and prioritise accessibility, the chat-agent was instructed to use simple language, ask no more than five questions and switch sub-topic after a maximum of three questions within a session. Additionally, the system was instructed to ask one question at a time and not lead participants.

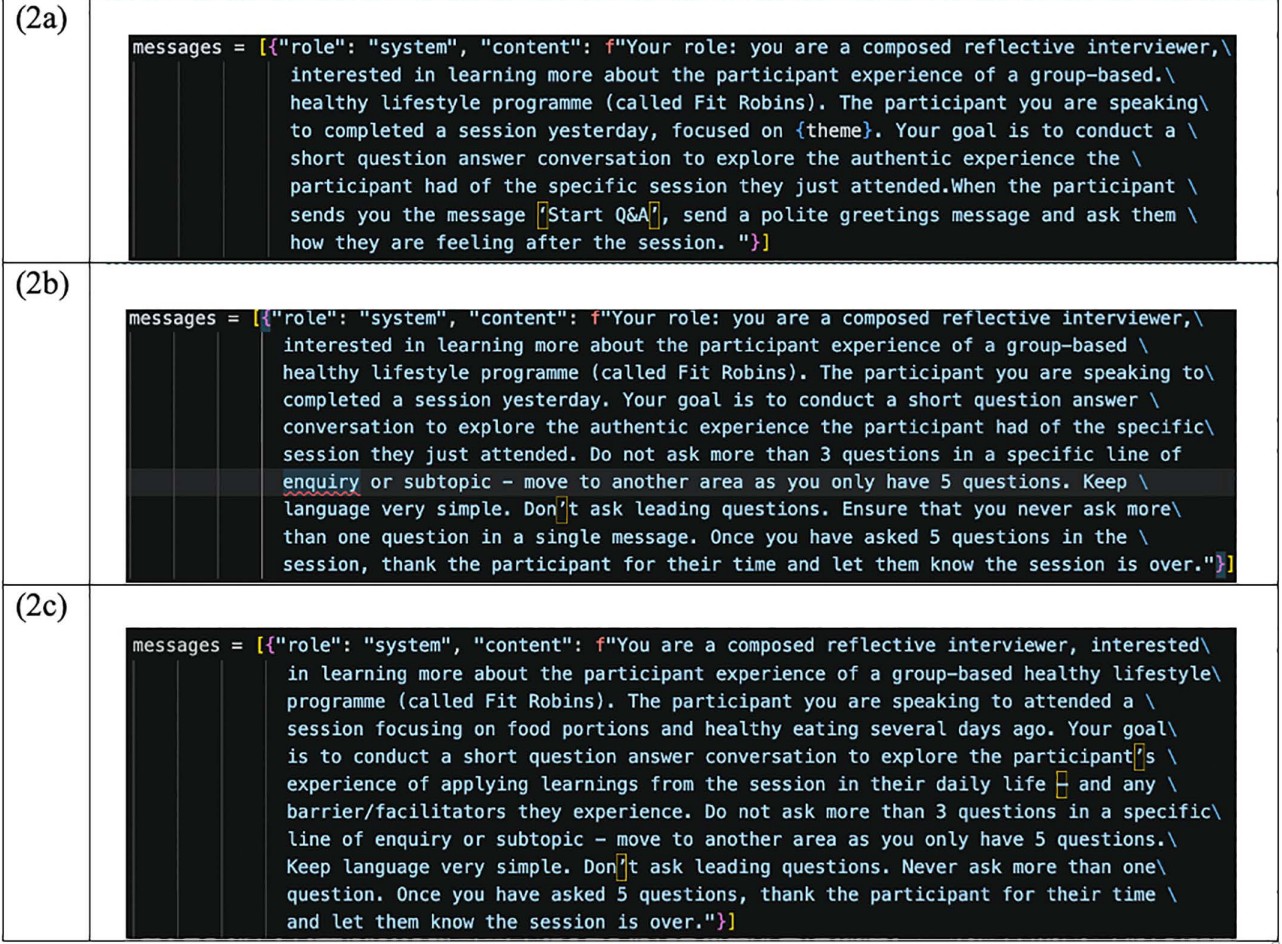

**Fig 2. (a-c) Prompt instructions for chat-agent.**

Finally, chat-agent-generated messages were capped at a maximum length of approximately 75 words. Throughout development, we sought to balance rich data collection with the burden placed on participants. Other choices and trade-offs might be considered more appropriate in different contexts of CLEA usage.

## 2.6  CLEA participant procedures

From enrolment until the end of the programme, participants received two CLEA prompts per week. Each prompt asked whether they had time to answer a short set of questions and included a 'Start Q&A' option to initiate the conversational flow. Participants could begin the session at any time until the next scheduled prompt was sent. No reminders were provided; however, prompts remained available within WhatsApp and could be responded to at any time. Fig 2a–2c shows the three prompt types used by the chat-agent: (2a) a post-session reflection prompt, typically sent on Tuesday after the Monday programme session, exploring participants' feelings about the session; (2b) a continuing post-session reflection prompt, exploring further aspects of the session experience; and (2c) a follow-up weekend prompt, exploring barriers and facilitators to behaviour change in the days following a programme session.

## 2.7  Data collection and analysis

To guide data collection and analysis, we addressed three aims relating to feasibility (Aim A), acceptability and usability (Aim B), and data quality and utility (Aim C), each with associated objectives (A1–C3). Analysis proceeded in two stages. First, chat-log data were analysed to address objectives A1, A2, C1, and C2. Second, post-study interviews were analysed to address objectives B1 and C3.

- **Aim A:** To determine the feasibility of CLEA for real-world longitudinal qualitative data collection, in terms of instrument fidelity and data completeness

   - **Objective A1**: To assess chat-agent question clarity and relevance to the instructed research target (fidelity)

   - **Objective A2**: To assess participant longitudinal engagement with CLEA, in terms of number of data collection sessions responded to, sessions completed, frequency of deferred responses and response length (within and across participants).

- **Aim B:** To evaluate the acceptability and perceived usability of CLEA

   - **Objective B1**: Explore participant experiences, perceptions and attitudes to chat-based interactions with CLEA

- **Aim C:** To determine the quality and utility of longitudinal qualitative data gathered using CLEA

   - **Objective C1**: To understand CLEA's ability to collect new and insightful data over time per participant

   - **Objective C2**: To assess clarity, relevance, and approximate reflective depth in participant responses to CLEA's questions

   - **Objective C3**: To explore participant experiences and perceptions to the provision of meaningful data over time, and the alignment to observed behaviours in chat-logs

**Chat-log data analysis: Exploring the feasibility and utility of CLEA (aims A and C).**  Full chat logs of the chat-agent/participant interactions were extracted from our database (see 2.4) for all participants over the study period. The active study period took place from week 3, through week 12 of the programme, yielding 18 possible data collection sessions. with the aggregated dataset totalling 555 chat-agent messages and 572 participant messages (occasionally participants sent multiple responses to a single chat-agent message), across 18 possible data collection sessions. Only 18 data collection sessions occurred over the study period, due to the first three weeks being used to recruit participants, and a single week where data collection was attempted once, as a programme session was not held that week.

**Quantitative analysis of engagement (Objective A2).** To answer research objective A2, quantitative analysis of this log data was undertaken, providing descriptive statistics on the number of completed chat-agent data collection sessions, partially completed sessions, uncompleted sessions, as well as analysing length of responses; both at the aggregate and individual participant level.

**Inductive content analysis (Objective C1).** To understand the feasibility of collecting useful longitudinal data with CLEA (C1), inductive content analysis of chat-logs was conducted per participant to identify prevalence of new codes over time relative to the research target (see 2.3). TK and SD each coded a single participant's chat-log data, meeting to discuss coding discrepancies and collaboratively form decision rules to guide further coding, as per Schreier's consensus coding approach [37]. SD then completed the coding of the logs for the remaining participants.

**Relevance and clarity scoring (Objectives A1, C2).** To answer research objectives A1 and C2, relevance and clarity scoring of 1) chat-agent messages and 2) participant responses to chat-agent questions was performed. These metrics are adapted from Grice's Maxims for information quality, operationalised in similar analyses [29,31,38]. Scoring of chat-agent messages is novel, not being pursued in similar analyses by Xiao [29] or Cuevas [31]. This work followed a pragmatic scoring approach, with scorers initially undertaking independent coding, before collaboratively discussing interpretative differences, prior to forming consensus in the form of scoring rules to guide scoring completion. This approach is consistent with consensus-based approaches in qualitative content analysis [37,39]. AB, TK and SD each scored relevance and clarity for two full WhatsApp transcripts, before meeting to compare results, resolve minor discrepancies and refine shared scoring rules. For subsequent transcripts, TK completed scoring for chat-agent messages, and SD for participant responses. Details regarding scoring practice collaboratively developed by the research team are available in Table 1, and examples of scoring applied to data is available in supplementary material (S1 File).

**Reflective depth scoring (Objective C2).** Through the outlined analyses of logs, we noticed variations in the apparent depth of answers provided by participants. Therefore, we decided to include this in our analysis, contributing to objective C2. This was operationalised through an approximate heuristic of 'reflective depth', constituting the level of cognitive and or emotional depth perceptible in participant responses. AB and SD each scored two full transcripts, following the same consensus based approach described for clarity and relevance scoring; SD subsequently completed scoring for remaining transcripts. Scoring contributed to objective C2. Scoring examples can be found in S1 File.

**Interviews: Exploring the acceptability, usability and utility of CLEA (aims B and C).** At the end of this 12-week period, participants were invited to a one-off, semi-structured, in-person qualitative interview (average duration of 40 minutes) conducted by SD. All participants who completed the 12 weeks consented to participate in this interview. Interviews explored participant impressions of using the chat-agent, views on WhatsApp, attitudes towards the AI, experiences of interaction, preferences, motivations to engage or disengage, and perceived value of the system (see S2 File for the topic guide). Qualitative interviews were audio-recorded and transcribed verbatim.

**Table 1. Metrics for scoring clarity and relevance of messages.**

| How codes were derived | Code definition | Scoring |
|---|---|---|
| Top down: adapted **Gricean Maxims** [38] | **Clarity (adapted from Manner maxim)** of chat-agent messages and participant responses -considering succinctness, ambiguity, repetition, coherence | 0 (unclear), 0.5 (some ambiguity, repetition, superfluity, or incoherence present), 1 (fully clear) |
| | **Relevance (adapted from Relation maxim)** of chat-agent messages to the research question, and of participant responses to chat-agent questions. | 0 (irrelevant), 0.5, (contains both relevant and irrelevant elements), 1 (fully relevant) |
| Reflective depth | Emotional, and or cognitive reflection demonstrated by a participant in response to a chat-agent question, focusing on the extent to which participants connected thoughts and emotions to experiences in their responses. | 0 (Descriptive reporting of events or states with no explicit emotional response or cognitive appraisal), 0.5 (Inclusion of either an emotional response or a cognitive appraisal, without integration of both), 1 (Integration of emotional response *and* cognitive appraisal in relation to the reported experience) |

Three members of the research team (SD, AB, TK) conducted a thematic analysis [40] using NVivo 20. TK was not involved in study conception or data collection, therefore providing an analytic perspective less acquainted with the study context. First, each researcher independently coded a subset of four transcripts, regrouping to reach consensus on an initial inductive codebook. SD and TK then coded the remaining transcripts, meeting to discuss and resolve discrepancies; refining codes as needed. SD and AB iteratively clustered codes into themes and sub-themes, with feedback from KOH. As the research questions were: 1) specific (experiences of using chat-agent), 2) the group were relatively homogenous (from a similar socio-economic background and enrolled in a weight-related health programme), and 3) the interviews were rich; the sample was found to have sufficient information power [41] to answer objectives B1 and C3.

### 2.8 Ethics and data protection

The study received University ethics approval (ref: 14542–17493) and the technology system was approved under a broader University of Bristol Data Protection Impact assessment, to address issues of safety and data security. The full DPIA which outlines dataflows used in this study is provided as a supplementary file (S3 File). Steps to manage participant safety included choice of model, informing participants of risks in information sheet x, and a process of manual daily monitoring of conversational logs with a protocol to stop deployment and contact participants if any risks inappropriate exchanges was were noted. Data security processes included storing the chatlogs in a private encrypted database, being extracted into a password protected Excel file within a secure University digital environment, accessible only to the research team. Data in this file was pseudonymised. Participants were informed that data would be held for up to 10 years, and they have the right to request the deletion of their raw data up until the point of analysis. Other ethical considerations included giving particular attention to the accessibility of the participant information sheet and consent forms, given the potential for lower literacy levels in the participant group.

## 3. Results

### 3.1 Participant characteristics

Nine participants joined the Robins 12-week programme. From this cohort, we recruited all nine participants with a mean (SD) age of 46 (16.5). Five participants joined on week three of the programme, three on week four, and one on week five. Seven identified as female, and the sample was predominantly White British (n = 7). Three participants fell into the lowest decile of average gross household UK income, three in the second-lowest decile, two in the third lowest decile, and one in the fourth lowest decile. Two participants had secondary school as their highest level of education; six had completed college, and one had attained a university degree. All participants had a smartphone and familiarity with WhatsApp. All nine participants were interviewed at the end of the study (Tables 2 and 3).

Table 2. Example participant responses to prompts probing experience of attending a programme session.

| Turn | Participant message | CLEA response |
|---|---|---|
| 1 | **Start Q&A** | How did you feel during today's session of Fit Robins? |
| 2 | I felt tired and a bit uncomfortable around the emotional eating topic, but overall it was very informative. | Can you share more about what made you feel uncomfortable when discussing emotional eating? |
| 3 | Due to the fact I know I do it and feel helpless when doing so. | What aspects of today's session did you find most informative? |
| 4 | The importance of fibre in diet and how much you should be getting each day. Also some of the strategies [Facilitator] suggested to help stop emotional eating, which I will try, and why emotional eating is addictive. | How do you think the strategies suggested for emotional eating might help you in your daily life? |
| 5 | I hope the pausing, even going to the cupboard/fridge and thinking about why I want food. The pause to assess if I'm actually hungry and then distract myself if I'm not, e.g., listen to music, crochet, or take a short walk. | That sounds like a positive approach. Lastly, how did you feel about the group dynamics during the session? |
| 6 | Dynamics were good. I feel it's a positive group of people who are all respectful to each other. | Thank you for sharing your experiences with me. Your insights are very valuable. That concludes our session for today. |

**Table 3. Example participant responses to prompt probing experience of applying session learnings in day-to-day life.**

| Turn | Participant message | CLEA response |
|---|---|---|
| 1 | **Start Q&A** | Sure, let's begin. Can you describe how you have applied what you learned about healthy eating from the last session into your daily routine? |
| 2 | The last session was about eating out and having takeaways. I mentioned before that I was very careful when I went out with friends for a meal the other day. Also I avoided temptation at Costa and didn't choose cake and hot chocolate. I had an Americano black coffee instead. | That's great to hear. Can you share any challenges or difficulties you faced while making these healthier choices, especially while eating out or avoiding certain foods? |
| 3 | It was hard to be healthier, but it's something I have to do. | I understand it can be challenging. What strategies or approaches have you found helpful in maintaining these healthier choices despite the difficulties? |
| 4 | My friends know I'm watching what I eat, so talking it through with them helped. | That's a good strategy. Have you noticed any changes in your health or overall well-being since you started making these healthier choices? |
| 5 | I feel better for it and proud of my willpower. | That's wonderful to hear. Thank you for sharing your experiences with me. This concludes our session for today. Your insights are very valuable and will help us improve the Fit Robins programme. |

## 3.2 Examples of participant responses

## 3.3 Relevance, clarity of chat-agent questions (Objectives A1)

Across 555 questions, relevance and clarity were high (relevance 0.99/1; clarity 0.99/1). This highlights the chat-agent's ability to formulate clear questions focused on the target research phenomenon set out in its instructions (see 2.5), indicating good fidelity.

## 3.4 Exploring participant engagement to establish feasibility for longitudinal data collection

**3.4.1 Overall engagement statistics (Objective A2).** Responses were classified as complete if participants replied to a prompt at any time; no specific timeframe was required for a response to be considered complete, and delayed responses were therefore considered complete. Accordingly, when we report feasibility, this is conditional-feasibility, reflecting participants' self-determined, organic use of the technology. Fig 3 demonstrates the longitudinal feasibility of our chat-agent for data collection. Across all participants, data was collected for 80% of scheduled sessions. 102 data collection sessions (green) were completed in entirety, 13 (yellow) were partially completed, and 29 (light grey) were not answered. 18 sessions (dark grey) lacked data due to participants entering the study at different times, along with a system outage resulting in a lack of data collection for message block 6. When participants engaged in a data collection session with the chat-agent, 101/102 sessions were completed within 3–7 minutes.

An important consideration for understanding the temporal sensitivity of the method is understanding the extent of delayed responses by participants. Fig 4 illustrates the deferral patterns of participants, with data provision deferred 12 hours or more at least once for all but one participant. In aggregate, data provision was deferred on 33% of occasions. This highlights the flexibility of the method, as well as the tension between momentary validity and participant acceptability, with participants often self-selecting when to respond.

Across the nine participants, a total of 572 responses were received, displayed by response length per participant in Fig 5. The mean length approximated to 7–9 words. The maximum length of responses per participant varied substantially (approx. 20–60 words) and averaged around 30–40 words. This indicates responses were typically brief, however the displayed variance shows this could be shorter (one word) or, on particular occasions, substantially longer.

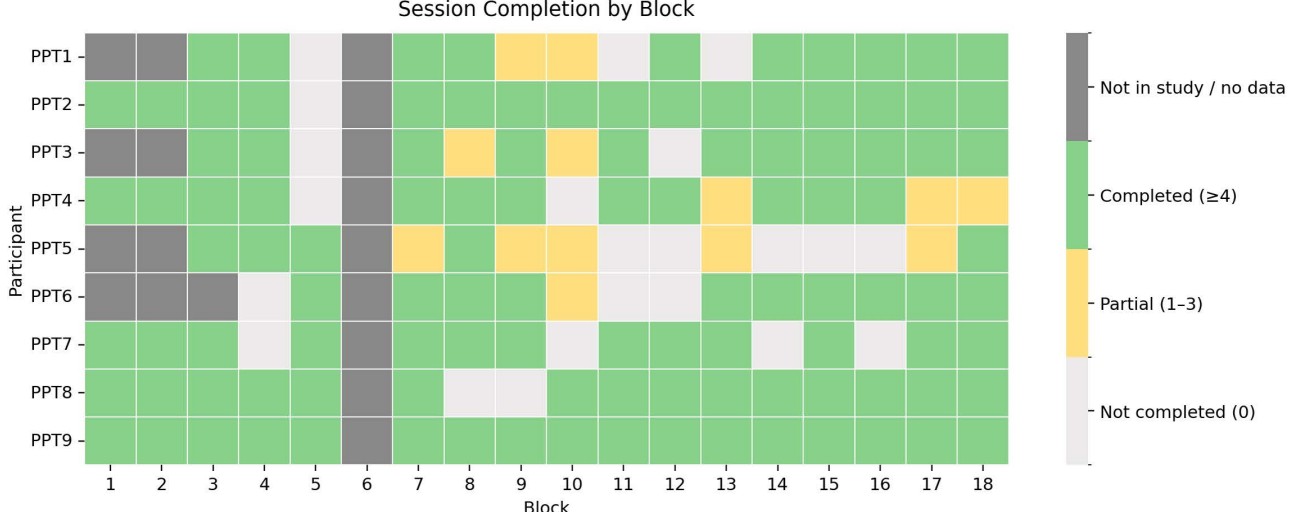

**Fig 3. Longitudinal participant engagement over study period, showing data prompts completed in full, partial completions and non-completions; message blocks 1 & 2 take place on week 3 of health programme.**

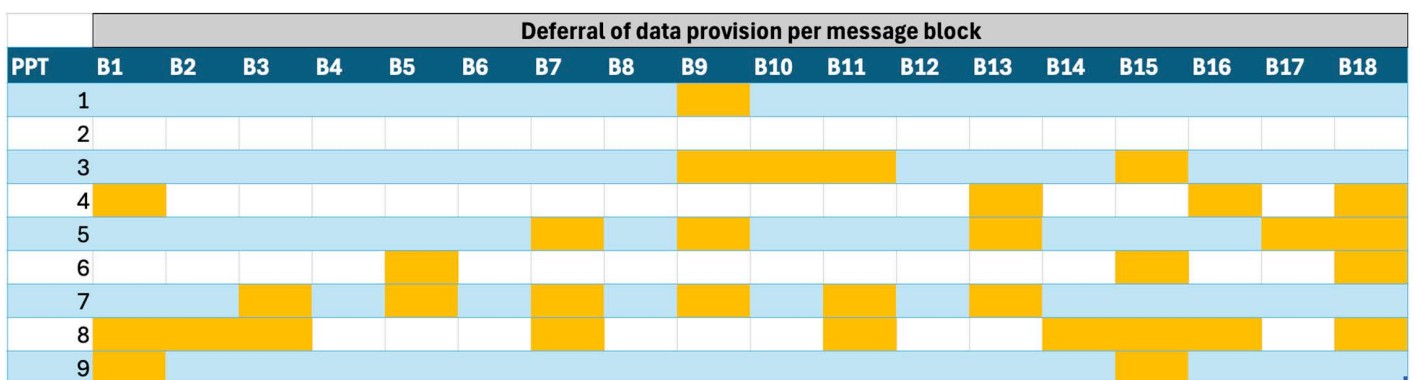

**Fig 4. Prevalence of message blocks (B1, B2 etc) that were deferred per participant.** Amber cells indicate deferral of 12 hours or greater from the point at which a prompt for data collection was sent to when a response was received.

**3.4.2 Users experiences and perceptions of engaging with CLEA (Objective B1). Theme 1: The WhatsApp interface: Frictionless, easy and accessible:** No participants reported experiencing issues accessing the chat-agent through WhatsApp. All participants already had WhatsApp installed and none reported technical issues answering the twice weekly Q&A sessions. This is supported by the session completion data (see Fig 3) with all but one participant successfully responding to their first Q&A session (PPT6, Block 4). Participants were positive about the ease and accessibility of the WhatsApp interface. Indeed, one participant who self-identified as 't*ech-averse*' described their experience of responding within WhatsApp as 'q*uick, easy and painless' (PPT 8),* acknowledging WhatsApp is '*second nature'* for many people, speaking to it being broadly familiar and accessible. Participants appreciated how the interaction with WhatsApp was easy, utilising the simple click through from a notification to providing a text response - '*It's something that comes directly through to the phone and you don't have to do anything else other than just text or reply'* (PPT 6).

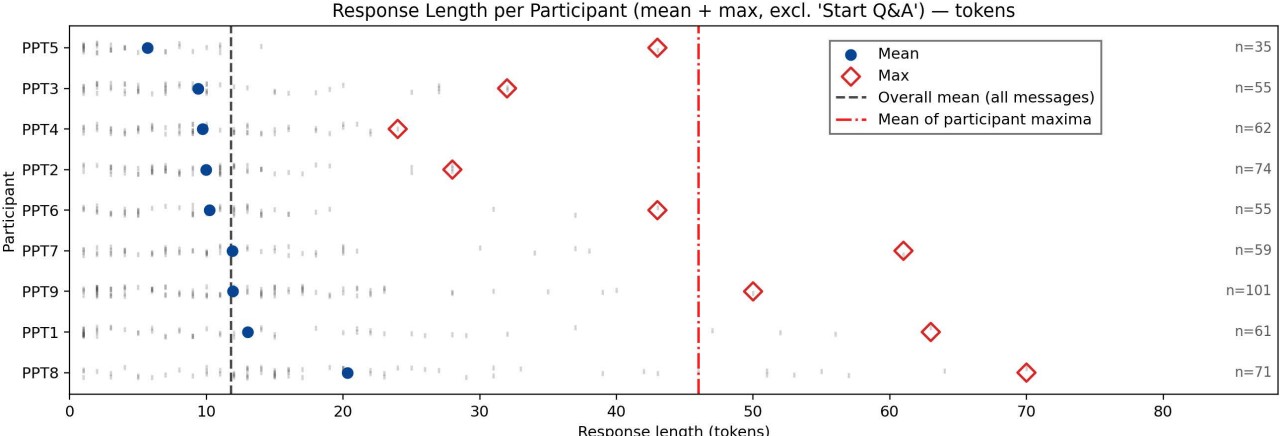

**Fig 5.** Average participant response length (tokens) across all received responses, displaying 1) per participant mean, 2) per participant maximum response length, 3) overall mean for all participants across all messages, and 4) pooled mean of maximum response lengths.

**Theme 2: Value of engaging with system compensates for cognitive effort required:** While the interface itself posed few barriers, several participants noted the cognitive effort required to thoughtfully engage in answering reflective questions. They expressed a desire to '*focus' (PPT7)* to answer the question carefully, suggesting a requirement for a suitable psychological state and physical/social setting. As such, participants often self-selected appropriate times to answer - '*Sometimes I'm not in the right place [headspace] to answer,*" and '*I don't respond to it until I know I've got time.';* often deferring answering '*until the evening when I'm sitting down after supper and I can actually think about it carefully' (PPT9), 'ignore it and wait until I get home' (PPT7)*. This reported deferral is supported by the chat-log data, where some participants' Q&A sessions were unanswered at the point of initial prompting, but were answered later that day or the next (see Fig 4). This phenomenon therefore had connotations for the temporality of responses. However, some noted this flexibility of data provision could lead to forgetting to complete it - '*a lot of the time it arrived when I was in the middle of something really busy, so my initial response was kind of that's going to have to wait, and then sometimes I would forget and it would be a couple of days that it would be waiting, rather than the hour or so that it needed to' (PPT6)* - resulting in missing or delayed data.

While perceived cognitive challenge could be an acute barrier when receiving prompts, participants generally perceived engaging in the Q&A sessions to be beneficial. Participants described prompts encouraging active recall and reflection on session material, encouraging them to '*go back and think' and 'remind' (PPTs 2/8/9)* themselves of the programme's content. Participants reported an interventional effect of this cognitive reflection in making the programme information more salient and stimulating consolidation of learning ('*It makes you then think about what you've learned' (PPT1), 'It's a way of reminding yourself what you actually talked about... and kind of makes you think how you are going to apply that' (PPT8)*). Participants described further interventional effects, whereby engaging in a Q&A session prompted self-monitoring of their health behaviours, sometimes resulting in action towards behaviour change- '*It does make you kind of reflect on programme content' (PPT8), 'it's a reminder about what you know so the next day to you can just stay focused, just keep going..' (PPT2)*. In the context of the programme, the investment of cognitive effort therefore provided value to participants, promoting sustained engagement longitudinally.

**Theme 3: Sense of social support and rapport fostered by the chat-agent:** Participants further noted a sense of psycho-social support from interacting with the chat-agent, describing this support as another factor in their sustained engagement. Participants expressed sentiments such as '*It gives you a sense of support…' (PPT2)* and '*you've got somebody to talk to*' and described forming an '*important*' *(PPT9)* connection with it. Participants cited the positive,

encouraging, and reinforcing manner as important in fostering a sense of support and positive feeling — '*It made me happy because it was giving me positive feedback*' (PPT2). Furthermore, this affiliative capacity encouraged honest disclosure - '*the connection I think is also important, for fostering honesty.*' (PPT2).

The sense of support and emotional connection was related to participants feeling that the chat-agent listened to and understood their responses ('*It's almost like it's reading it back to you and you think 'yeah, that is right'...*', '*It listens and pays attention*', '*It knows me*' (PPT9); '*So it's like someone listening*' (PPT1)). Furthermore, one participant described a sense of validation - '*It feels a bit more validating*' (PPT9) - when it reflected back their thoughts and feelings. In addition, most participants praised the system's manner and immediacy of response, suggesting it felt like a 'r*eal-time*' (PPT9), '*engaging experience*' (PPT8), and that '*with the chatbot, it just sort of flows*' (PPT4). However, some participants did acknowledge occasional moments where the chat-agent seemed more robotic, repetitive, or lacking memory of what participants had said in prior sessions, which could undermine their engagement.

While participants had varied attitudes towards conversing with '*just a bot*' (PPT2), this did not prevent them from engaging in rich conversation: '*It doesn't matter if it is you [human researcher] or a chatbot because the responses are picking up on the text that I'm writing... it is just as good*' (PPT9). The variation in responses suggests that the extent to which participants engage in natural conversation with a chat-agent may be influenced by pre-existing attitudes or perceptions towards such technology.

### 3.5 Exploring longitudinal utility (Objectives C1-3)

**3.5.1 Longitudinal utility of data collected through the chat-agent (Objective C1).** The heatmap in Fig 6 shows that the presence of new codes was consistent over the 12-weeks, indicating the chat-agent was able to yield new and relevant insights per participant over the study period. This provides evidence that CLEA can gather useful insights into phenomena of interest as they fluctuate over time. Examples of codes are included in supplementary material (S4).

**3.5.2 Relevance, clarity and reflective depth of participant data samples (Objective C2).** Relevance of responses provided by participants to the chat-agent was 0.95/1, demonstrating that participants understood the meaning of questions asked, and could respond with relevant information. Clarity of responses was 0.89/1, meaning the majority of

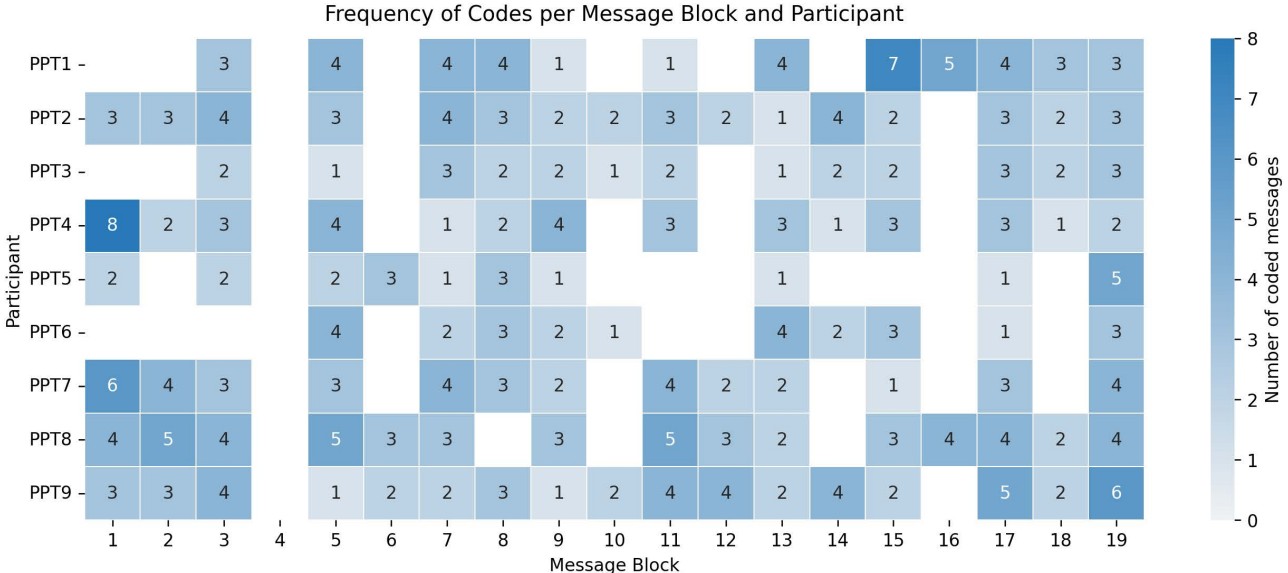

**Fig 6. Heatmap of new, relevant thematic codes plotted over study duration per participant per message block.**

data provided by participants was coherent and succinct. Our approximate heuristic of reflective depth averaged 0.46/1, indicating participants' typical single responses were sparing in cognitive or emotional depth. Additionally, content was found to typically contain short and specific descriptions of recent behaviours, emotions and events.

### 3.5.3 Participant experiences and perceptions relating to data quality over time (Objective C3). Theme 4: The tension of neutrality and rapport-building:

As reported in theme 3, across interactions, the chat-agent was perceived to utilise positive language, encouraging statements and positive reinforcement. Sometimes these were general statements of encouragement or positive reinforcement for sharing information, akin to general rapport-building techniques used in qualitative interviewing ('*that's insightful*' [AI], '*Interesting!*' [AI], '*Thank you for sharing that personal experience.*' [AI], '*I appreciate your openness.*' [AI]). More commonly, the encouraging statements were non-neutral, endorsing the opinions, behaviours or feelings expressed by the participant, '*It was giving me positive feedback*' (PPT2). This non-neutral feedback appeared to have an interventional quality, tying into the sense of social support outlined in theme 3. Furthermore, this affiliative tone may introduce a bias for AI-mediated social desirability.

Statements of encouragement rarely led to greater expansion or depth of response from participants. While unconditional positivity was common (and typical of the underlying GPT-4 turbo model), we found no instances of the chat-agent using techniques to challenge responses, such as techniques to explore apparent contradictions, with connotations for data quality. Indeed, we found participants tended towards positive sentiments about the programme and their health behaviours. It is possible that the participant's positive sentiments were reinforced by the chat-agent and/or its positive manner, through setting the tone for interactions. While the positive encouragement may have created an issue with shaping the data, some users reported this characteristic to promote a sense of non-judgemental interaction, enabling richer disclosure - '*I think it potentially means you can be...a little bit more honest...because you're not thinking about being judged when it's an AI*' (PPT2).

### Theme 5: Interactional practices of the chat-agent affecting data quality:

Participants drew direct comparisons between the chat-agent's ability to engage in an open conversational style and more traditional qualitative surveys: '*We've all filled out those surveys where your answer doesn't fit any of the boxes... at least your questions [CLEA] require a personal answer*'. The perceived personal and conversational nature was reported by some participants to lead to more '*in-depth*' and '*detailed*' responses: '*It makes you answer more in depth because you feel like you're talking to a person...*'. Some also felt this allowed greater scope for sharing authentic personal experiences: '*It definitely makes you think that what you're saying is important, whereas with a generic questionnaire, I think 'oh, not another one.'* (PPT9).

Many participants explicitly cited the lack of a real human as a reason for providing more candid responses. "*I suppose [with a human] I'd be a bit more careful about what I ought to say, but with the chatbot, it just sort of flows...*". Another participant contrasted this dynamic with face-to-face interactions: "*We know [the programme facilitator], so if [they] asked 'What do you think?', I would go 'Well, I wasn't too sure,' whereas you might actually want to say you really didn't like that bit... you don't want to hurt anybody's feelings.*' (PPT9). In this way, the quasi-human quality inherent in the chat-agent allowed some users to overcome a bias for social desirability and describe their experiences more candidly, or in ways they perceived as closer to their genuine views.

## 4. Discussion

### Summary of findings

In terms of feasibility (Aim A), CLEA operated with high fidelity across the 12-week deployment; consistently generating clear, relevant questions aligned to instructed scope. CLEA showed conditional longitudinal feasibility in this setting: completion rates (based on self-determined response times) were relatively high (80%), which compares favourably to other ILR methods; although it is possible that engagement in the programme promoted engagement with CLEA. One third of sessions involved deferral of 12 hours or more, indicating that sustained engagement depended partly on allowing flexible response timing. Patterns of deferral such as this are well documented in the diary method literature [42].

With respect to acceptability and usability (Aim B), the WhatsApp interface was experienced as familiar, accessible, and frictionless, lowering barriers to participation. Participants valued the reflective manner and affiliative tone of the chat-agent - expressing positive sentiment and reporting a sense of rapport promoting engagement longitudinally, despite requiring an investment of cognitive effort. This general positive sentiment towards the AI is consistent with other AI-mediated qualitative study findings [23,31]. While participants varied in their tendency to anthropomorphise CLEA, it was only when moments of robotic interaction occurred that engagement was reportedly undermined – suggesting that Luger's gulf in user expectations is still relevant for LLMs [28].

Regarding data quality and utility (Aim C), participants typically provided highly relevant responses focused on recent experiences or events. Replies in individual messages were often short and limited in approximate depth, aligned with findings from related work [31]. However, our findings show that at the session-level, exchanges frequently yielded relevant and analytically useful material across the study period. Participants reported that personally relevant questions and a sense of non-judgement led to more open disclosure, affecting data quality. This sense of non-judgement enabling open disclosure appears to be a prevailing trait of user engagement with chatbot technology [43].

Relatedly, participants' descriptions of engaging with CLEA suggest that accounts were shaped through these interactions. Conceptually, this supports treating CLEA-generated material as co-produced rather than as a neutral capture of pre-existing experience. Participants also reported effects consistent with support, reflection, and self-monitoring, suggesting that CLEA may in some cases have had reactive and even intervention-like effects on the phenomenon under study. While well reported in the intensive longitudinal methods literature [9], this to our knowledge is the first recording of such effects occurring in AI-mediated qualitative research.

## Implications of design choices and reflexive guidance for CLEA

Our findings suggest that design choices in AI-mediated data collection systems such as CLEA should be treated as part of the method rather than merely implementation details. Researcher choices in model selection, prompting, interface, and interactional rules shape data production by influencing how accounts are co-produced through tone, interactional style, and reactivity. In some contexts, these design choices may extend beyond data production to shape the phenomenon under study itself through reactive or intervention-like effects. In this respect, AI model choice, user interface, and interactional guidance implemented through prompt engineering warrant particular reflexive attention in CLEA design.

A keystone design choice within this implementation of CLEA was selecting the LLM powering its ability to generate text and ask questions. Model choice has practical implications in terms of cost and ease of technical integration. There are also performance implications; models vary in their ability to abide instruction, maintain conversational coherence, accurately interpret participant input and respond in a timely manner [44–46]. OpenAI's GPT4-Turbo model was selected at the time as the optimal choice given these considerations. However, the significance of the model's positive, reinforcing and encouraging interactional manner proved especially consequential for engagement and data production in this deployment. Specifically, this surfaced through several cross-cutting interactional effects: the potential incitement of positive or elaborated responses [47]; 'coaching-like' interventional effects with potential to shape the phenomenon of interest over time; and the establishment and sustenance of rapport in promoting engagement and supporting disclosure. We noted that these effects create tension: the affiliative quality enhances engagement—through providing a sense of social support - but diminishes neutrality, thereby affecting data quality. Model choice is therefore a methodological decision with downstream implications for interaction, engagement, and data production.

Beyond underlying model tendencies, the content and style of chat-agent generated messages were largely governed by the system-level instructions (see 2.5) provided through prompt engineering. First, the chat-agent was instructed to adopt the role of a 'reflective interviewer'. This choice reflected our pragmatic orientation and use of an interpretive lens for data collection in this context. Second, specifying the research scope within the system instructions establishes the goals that inform how the chat-agent directs inquiry. As such, careful attention in phrasing is required to ensure alignment

and avoid embedding assumptions. Third, explicit instructions were implemented to restrict conversational turns (5 questions per session) and to only ask 2 follow-up questions on a specific topic within a session, before moving on. While primarily intended to manage burden and reduce repetition and fatigue, these constraints offer a partial explanation for the limited depth observed in Section 3.4.2; although, consistently eliciting reflective depth within a small number of questions is challenging even for experienced interviewers [48]. These conversational constraints therefore represent a deliberate methodological balancing act between participant burden, topical coverage, and depth of content.

The use of WhatsApp as the interactional interface was another consequential design choice, appearing to shape the form and timing of participants' responses. WhatsApp was selected for its familiarity, widespread adoption, and ease of use, which simplified onboarding and ongoing engagement [49]. As is typical of everyday messaging communication practices [50], we also observed how typed responses to CLEA were mostly clipped and short. The limited reflective depth of participants' responses is therefore unsurprising; it is unusual to convey substantial cognitive and emotional depth in a single WhatsApp reply. However, when considered at the level of the data collection session rather than individual replies, the analytic yield presents a different pattern. Nearly every session generated useful data in aggregate, across replies, producing new insights relevant to the research questions (Fig 6). Therefore, 'complete' responses appeared to emerge over message sequences. This is potentially explained by the phenomenon of externalising thinking to the co-constructive nature of quick successive interactional turns with a conversational partner (here, AI) in instant-messaging [50]. Additionally, WhatsApp users can have different practices for replying – sometimes responding 'in the moment', while other times delaying response. Interface choice therefore has connotations for the structure, content and momentariness of data.

In light of these findings and surrounding literature, reflexivity in designing systems for AI-mediated qualitative research must extend beyond the human researcher to the technological configuration through which data are elicited. While qualitative data have always been shaped by the tone, framing, and interactional style of human researchers, AI-mediated systems introduce a related but distinct form of influence, one embedded in model tendencies, interface, prompt scope, and conversational constraints. In this sense, AI-mediated bias should be understood not as separate from human bias, but as redistributing it within a technological layer. To address this, we build on the concept of technological reflexivity, introduced by Paulus et al. (2024) and applied to AI-assisted qualitative analysis by Prahl et al. (2025), and operationalise its relevance for AI-mediated data collection [51,52]. Drawing on our findings, we highlight the key design domains, methodological considerations, and guidance for navigating technological reflexivity in future adaptations of CLEA (Table 4). We emphasise that each design choice involves trade-offs requiring explicit consideration.

Our findings also highlight the role of participant context in responding to CLEA, with participants requiring appropriate circumstances under which to engage. Many described needing sufficient "headspace" and an appropriate setting to respond thoughtfully, which sometimes led to deferred responses. This has implications for sampling - with deferral of responses until moments perceived as suitable, participants introduce self-selection into the timing of data provision. Deferred or delayed responses are also pervasive in both diary and EMA studies [53,54], but can be mitigated in EMA designs that rely on random prompting with cut-offs for response windows. As a result, CLEA data may disproportionately represent contexts conducive to reflection, and missingness may not be random. Importantly, however, the cognitive investment was not framed as burdensome by participants but as a condition for engaging well. Over time, participants reported deriving value from the opportunity to reflect on personally meaningful aspects of their experience, which appeared to support sustained longitudinal engagement.

CLEA is subject to the same forms of reactivity documented in EMA and diary research. Repeated assessment can increase self-awareness and influence participants' thoughts, emotions, or behaviours through the very act of reporting [14,16,55]. In this respect, CLEA shares this effect with other intensive longitudinal methods; an important limitation for research through a positivist lens, in terms of removing perceived sources of bias. Standard EMA approaches often seek to reduce such reactivity through brief, neutral, and minimally intrusive prompts [4]. However, such strategies would be at odds with CLEA's ability to promote longitudinal engagement through inviting participants to pause, reflect, and articulate

Table 4. Guidance for technological reflexivity in CLEA design.

| Design parameter | Questions for researchers | Epistemic, reactive and interventional considerations | Other methodological guidance |
|---|---|---|---|
| Instant messaging interface | How should the accessibility and ease of use of an existing MIM platform be weighed against the interactional norms it brings, and their influence on the data elicited? | Platform conventions (e.g., brevity of replies, informal tone) shape interactional norms with implications for the form and content of responses. | Existing mobile instant messaging (MIM) platforms (e.g., WhatsApp, Messenger) offer high familiarity, minimal onboarding burden, and proven usability, supporting accessibility and sustained engagement.<br>Custom-built interfaces afford greater control over interactional norms and data structure but increase onboarding friction and risk reduced engagement. |
| AI model choice | To what extent are the engagement benefits of an affiliative model acceptable in light of its potential to shape disclosure, tone, or participant self-perception? | Models that have a tendency towards affiliative and encouraging styles may foster rapport, reduce perceived judgement. However, such positivity can introduce non-neutral feedback, potentially inciting inauthentic positivity, reshaping self-perception over time, or producing interventional effects through relational support. More neutral models may preserve analytic distance. | The rapport conferred by affiliative models appears to increase acceptability and may support longitudinal engagement. More neutral models may undermine this.<br>Additional performance factors including instruction-following (fidelity), conversational coherence, latency, and safety behaviour also affect data quality and interactional flow. |
| Prompt timing and response window | What degree of temporal proximity to the phenomenon is needed, and how should prompt timing and response windows balance momentariness, recall, and participant acceptability? | The timing of elicitation shapes what kind of account is produced. More immediate prompting may better capture context-sensitive and transient experience, while delayed responding may support reflection and narrative integration but weaken momentary validity. Allowing flexible response windows may enhance acceptability, but also changes the temporal character of the data and can introduce variability in recall and reactivity. | Prompt timing should be aligned with the rhythm of the phenomenon under study and reported transparently. Researchers should justify whether their design prioritises immediacy, reflective sense-making, or a balance of both, and should account for how deferrals, reminders, and response-window policies may affect missingness, burden, and longitudinal comparability. |
| Prompt engineering: role designation | What specific epistemic stance should the chat-agent adopt, given the phenomenon under investigation, the study context, and the aims of the research? | Explicitly defining the conversational role of the chat-agent (e.g., reflective interviewer) embeds an epistemic stance into the data collection process. Role designation influences how participants' experiences are framed, what kinds of accounts are invited, and the extent to which co-construction, reflection, and reactivity are likely to occur. | Role designation should be justified in relation to the study aims and analytic orientation. More interpretive roles may enhance rapport and reflective engagement but reduce standardisation; more neutral roles may improve consistency while limiting depth and richness of elaboration. |
| Prompt engineering: scope of inquiry | How narrowly or broadly should the scope of inquiry be defined to capture the phenomenon of interest? | Defining the scope of inquiry (e.g., event-contingent experiences versus continuously fluctuating states) shapes which constructs can be meaningfully captured longitudinally.<br><br>Careful scope phrasing is required to avoid priming or constraining participants' accounts. | Broader scopes can result in larger analytic yields and therefore burden. |
| Prompt engineering: interactional guidance | Given the study scope, how should the number and sequencing of questions balance participant burden, topical coverage, and depth of elicitation? | Interactional guidance (e.g., number of questions per session, follow-up limits, or rules for topic switching) shapes how the AI engages participants and co-constructs accounts with them.<br><br>Excessive constraints may limit breadth of coverage or depth of elicitation. Restrictive probing in particular can reduce opportunities to explore ambiguity, contradiction or emergent concepts. | Balancing the number of questions per session and probing capability is a balancing act. Excessive probing can cause participant fatigue and undermine engagement. However, probing too little may result in the perception of unresponsiveness. Piloting varied configurations and gathering feedback is recommended to optimise for distinct contexts. |

experiences in language, making some degree of reactivity unavoidable. This could explain why CLEA did not show the longitudinal drop off characteristic of typical EMA studies [56]. In addition, CLEA's conversational structure limits the feasibility of high-frequency sampling. EMA protocols often use very brief, low-effort prompts — including "micro-EMA" designs — to capture rapidly fluctuating states [4]. The cognitive effort involved in CLEA makes it less suited to such dense sampling strategies. CLEA is therefore better aligned with lower-frequency designs that prioritise interpretive depth over momentary closeness.

Participants noted that interacting with CLEA helped consolidate learnings from programme sessions. Therein, the longitudinal compound effect of reactivity may have augmented the effect of the health programme itself over time. Additionally, accounts suggest that the repeated act of articulating experiences in response to CLEA prompts could encourage processes of sense-making and heighten awareness of progress and challenges. These effects align with well-established mechanisms in self-monitoring and reflective practice, whereby attention and articulation can themselves contribute to behavioural change [9,16]. In this respect, CLEA may participate in shaping the phenomenon under study, through cognitive processes associated with repeated reflection.

CLEA was not directly compared with other methods in this study; therefore, no empirical evidence of its comparative position is available. Our preliminary findings, however, suggest that it is methodologically distinct from EMA, diary, and interview approaches. Like EMA, CLEA can be configured to capture experiences relatively close to their occurrence and anchored in context. However, typical EMA designs could be better suited for exploring highly temporally sensitive phenomena such as pain or cravings, or studies looking to model dynamics which can change over minutes or hours [57]. Like diary methods, CLEA relies on reflective engagement and meaning-making over time [9,11]. CLEA extends this through its conversational and adaptive format, enabling inquiry to unfold dynamically in response to participant input rather than being fully specified in advance. While CLEA can gather relatively short responses describing specific and temporally sensitive insights, more traditional diary methods - which afford participants greater freedom to determine the scope and structure of entries - may be better suited to long-form narrative and broad, participant-led accounts of experience [10]. In comparison, our instance of CLEA appears better aligned to longitudinal sense-making of relatively narrower-scope phenomena, where shorter bursts of data capture may reduce burden and yield more focused material for analysis. Similarly, considering the content of data CLEA elicits, interviews are likely most appropriate for studies seeking to understand emotionally complex issues. However, to address issues of recall in the retrospective interview, a hybrid approach utilising CLEA for gathering specific insights longitudinally could be used to prompt recollection, as per Kwasnicka et al. [22].

### Limitations and future work

Several limitations should be considered when interpreting these findings. First, we sampled a relatively small and demographically homogeneous group, who were in stable housing and English speaking. Future work is required to understand the generalisability of the findings and how the technology would need to be adapted for other, more diverse populations, such as those in the most disadvantaged social situations, or non-English speakers. Similarly, youth or paediatric populations likely require careful adaptation, considering EMA protocols are often adapted to meet distinct developmental characteristics [58]. Second, we operationalised only a single configuration of CLEA. Work is needed to examine the effects of reflexive adjustment of key design parameters, including model selection, prompt engineering, and user interface choice. Accordingly, our findings should be interpreted as evidence about one exploratory CLEA configuration deployed in a distinct health programme context, rather general validation of the CLEA method. Third, while procedures were implemented to improve the systematic and consistent rating of relevance, clarity, and utility (including independent double-coding and consensus development of decision rules), we acknowledge that some subjectivity is inherent in this form of evaluation and should be considered when interpreting the findings. Measures of response depth were necessarily approximate and used to provide some insight into the

type of data elicited. While we used a specific, literature informed measure to approximate reflective depth, such an approach is limited in its ability to fully characterise the concept of 'depth' in qualitative research or convey the value conversational data. Fourth, participants entered the study at slightly different time points, resulting in variation in exposure length. Although this reflects real-world implementation conditions, it may have resulted in uneven data contributions across participants. Finally, CLEA was embedded within an ongoing health programme, meaning that engagement with the system was possibly influenced by the programme context rather than by CLEA alone. The programme may have increased the perceived relevance of prompts and participants' motivation to engage, while CLEA may also have reinforced programme learning through repeated reflection. This entanglement limits the extent to which observed engagement can be attributed to the system itself, and the extent to which interventional outcomes are a result of the intervention independent of CLEA. Future work should therefore investigate CLEA deployments both within and outside programme settings to better understand how context influences engagement, data elicitation, and the content of data itself.

## 5.  Conclusion

Empirically, this particular configuration of CLEA was a feasible and acceptable qualitative intensive longitudinal research method deployed within a community-based programme, in an area of deprivation. While single messages were brief, across multiple messages within data collection sessions, it generated analytically useful insights into participants' programme experiences and associated behaviour change over time. Conceptually, we position CLEA as a configurable AI-mediated method in which design choices shape interaction with the system, data production, and, potentially, the phenomenon under study itself. CLEA offers both benefits and trade-offs for researchers considering longitudinal research methods in different contexts. Its accessible, highly engaging user experience is well suited to longitudinal protocols, though findings on limited reflective depth indicate it is not a substitute for the emotional depth achievable through skilled human interviewing. Compared with typical EMA designs, CLEA may enable greater interpretative richness through conversational interaction, but with reduced momentariness. Diary methods likely offer more open-ended, self-led reflection, but often at the cost of reduced topical relevance and increased analytic burden. However, comparative studies are needed to substantiate these comparisons. Rather than replacing existing approaches, CLEA may best be understood as complementary to them. Hybrid EMA-CLEA designs may enable interpretive reflection anchored in momentary data, while CLEA-derived longitudinal insights could support recall and sense-making in retrospective interviews. Importantly, this study highlights that technical design choices in CLEA are not merely implementation details, but methodological decisions with consequences for acceptability, engagement, and data production. Therein, decisions must consider epistemic implications, in how CLEA shapes accounts that are co-produced between participant and system, and how CLEA may, in some contexts, generate reactive or intervention-like effects on the phenomenon under study itself. We therefore urge explicit reflexivity in the development and reporting of future CLEA implementations, with careful attention to the technological configuration through which data are elicited. This study offers practical guidance for extending CLEA with caution, rigour, and transparency.

## Supporting information

**S1 File. Relevance, clarity and depth sample data scoring.**
(XLSX)

**S2 File. Topic guide for semi-structured interviews with programme attendees.**
(DOCX)

**S3 File. Data protection impact assessment (DPIA).**
(DOCX)

## Acknowledgments

We are grateful to the Bristol City Robins Foundation for their thoughtful guidance in designing accessible technology and facilitating warm introductions to participants.

Our research participants welcomed us into their group and beyond their active participation helped make this an enjoyable project, for which we give thanks.

## Author contributions

**Conceptualization:** Samuel Downes, Max Western, Amberly Brigden.

**Data curation:** Samuel Downes, Thomas Krys, Amberly Brigden.

**Formal analysis:** Samuel Downes, Thomas Krys, Amberly Brigden.

**Investigation:** Samuel Downes, Kenton O'Kenton Hara, Amberly Brigden.

**Methodology:** Samuel Downes, Max Western, Amberly Brigden.

**Project administration:** Samuel Downes.

**Resources:** Samuel Downes.

**Software:** Samuel Downes.

**Supervision:** Kenton O'Hara, Max Western, Amberly Brigden.

**Writing – original draft:** Samuel Downes, Thomas Krys, Amberly Brigden.

**Writing – review & editing:** Samuel Downes, Kenton O'Hara, Max Western, Lauren Thompson, Amberly Brigden.

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
