## [Decision Letter · Decision Letter 0]

24 Mar 2026

PDIG-D-26-00092Conversational, Longitudinal, Ecological Assessment (CLEA): Exploring a new AI-driven method for qualitative data collection in a behavioural health contextPLOS Digital Health Dear Dr. Downes, Thank you for submitting your manuscript to PLOS Digital Health. After careful consideration, we feel that it has merit but does not fully meet PLOS Digital Health's publication criteria as it currently stands. Therefore, we invite you to submit a revised version of the manuscript that addresses the points raised during the review process. Please submit your revised manuscript by May 23 2026 11:59PM. If you will need more time than this to complete your revisions, please reply to this message or contact the journal office at digitalhealth@plos.org.  Please include the following items when submitting your revised manuscript:* A letter that responds to each point raised by the editor and reviewer(s). You should upload this letter as a separate file labeled 'Response to Reviewers'. This file does not need to include responses to any formatting updates and technical items listed in the 'Journal Requirements' section below.* A marked-up copy of your manuscript that highlights changes made to the original version. You should upload this as a separate file labeled 'Revised Manuscript with Track Changes'.* An unmarked version of your revised paper without tracked changes. You should upload this as a separate file labeled 'Manuscript'. If you would like to make changes to your financial disclosure, competing interests statement, or data availability statement, please make these updates within the submission form at the time of resubmission. Guidelines for resubmitting your figure files are available below the reviewer comments at the end of this letter. We look forward to receiving your revised manuscript. Kind regards, Sulaf Assi, PhDAcademic EditorPLOS Digital Health Sulaf AssiAcademic EditorPLOS Digital Health Leo Anthony CeliEditor-in-ChiefPLOS Digital Healthorcid.org/0000-0001-6712-6626 **Journal Requirements:**

a. Please clarify all sources of funding (financial or material support) for your study. List the grants (with grant number) or organizations (with url) that supported your study, including funding received from your institution.

b. State the initials, alongside each funding source, of each author to receive each grant.

c. State what role the funders took in the study. If the funders had no role in your study, please state: “The funders had no role in study design, data collection and analysis, decision to publish, or preparation of the manuscript.”

d. If any authors received a salary from any of your funders, please state which authors and which funders.

2. We ask that a manuscript source file is provided at Revision. Please upload your manuscript file as a .doc, .docx, .rtf or .tex.

3. Please provide separate figure files in .tif or .eps format.

4. We have noticed that you have uploaded Supporting Information files, but you have not included a list of legends. Please add a full list of legends for your Supporting Information files after the references list.

5. Some material included in your submission may be copyrighted. According to PLOS’s copyright policy, authors who use figures or other material (e.g., graphics, clipart, maps) from another author or copyright holder must demonstrate or obtain permission to publish this material under the Creative Commons Attribution 4.0 International (CC BY 4.0) License used by PLOS journals. Please closely review the details of PLOS’s copyright requirements here: PLOS Licenses and Copyright. If you need to request permissions from a copyright holder, you may use PLOS's Copyright Content Permission form.

Potential Copyright Issues:

Figure1_SD.png :Please confirm (a) that you are the photographer; or (b) provide written permission from the photographer to publish the photo(s) under our CC-BY 4.0 license.

 If the reviewer comments include a recommendation to cite specific previously published works, please review and evaluate these publications to determine whether they are relevant and should be cited. There is no requirement to cite these works unless the editor has indicated otherwise.  **Additional Editor Comments (if provided):****Reviewers' Comments:** Reviewer's Responses to Questions

**Comments to the Author**

1. Does this manuscript meet PLOS Digital Health’s publication criteria? Is the manuscript technically sound, and do the data support the conclusions? The manuscript must describe methodologically and ethically rigorous research with conclusions that are appropriately drawn based on the data presented.

Reviewer #1: Yes

Reviewer #2: Yes

Reviewer #3: Yes

Reviewer #4: Partly

2. Has the statistical analysis been performed appropriately and rigorously?

Reviewer #1: Yes

Reviewer #2: N/A

Reviewer #3: N/A

Reviewer #4: I don't know

3. Have the authors made all data underlying the findings in their manuscript fully available (please refer to the Data Availability Statement at the start of the manuscript PDF file)?

Reviewer #1: No

Reviewer #2: Yes

Reviewer #3: Yes

Reviewer #4: No

4. Is the manuscript presented in an intelligible fashion and written in standard English?

Reviewer #1: Yes

Reviewer #2: Yes

Reviewer #3: Yes

Reviewer #4: Yes

5. Review Comments to the Author

Reviewer #1: Thank you for the opportunity to review this thoughtful and timely manuscript. The study addresses a highly relevant methodological question in digital health research: how conversational AI can be deployed for longitudinal qualitative data collection in real-world behavioral health settings. The manuscript is well structured, grounded in relevant literature, and reflexive regarding technological design choices.

That said, several areas require clarification and strengthening before publication. My review is uploaded as an attachment

Reviewer #2: This study offers an important conceptual contribution and demonstrates the potential of AI-mediated longitudinal qualitative research. However, in its current form, the epistemic positioning requires clarification.

1. While CLEA is acknowledged to not only collect data but also transform the studied process through reactivity and compound intervention effects, this needs to be more explicitly theorized. The distinction between programme effects and CLEA’s reflexive intervention should be clearly articulated; as it stands, the boundary between “measurement” and “accompanying intervention” remains ambiguous.

2. Although the co-constructed nature of the data is implied, its ontological and methodological implications require systematic treatment. In particular, the role of technological design as an epistemic actor should be placed within a more explicit theoretical framework.

3. The influence of model choice, prompt structure, and interface design on the nature of the data has not been empirically tested. This introduces the risk that findings are specific to the particular configuration used. Without systematic variation of these design parameters—or at minimum a stronger analytical justification—the generalizability and internal consistency of results appear limited. The study needs to make its conceptual and methodological framing more transparent, theoretically grounded, and robust. Such reinforcement would elevate the work into a reference-worthy contribution for the field.

4. In addition, the discussion should consider AI-mediated guidance in relation to human researcher influence. In qualitative research, data is inherently co-produced between participant and researcher; the interviewer’s tone, framing, and interaction shape responses. AI introduces a parallel, though distinct, type of influence: more systematic, scalable, and embedded in design parameters. The study should explicitly theorize similarities and differences between “technological bias” and “human bias” to clarify the epistemic stakes. Otherwise, AI’s effect may be over- or under-problematized, and the human role implicitly assumed neutral, leaving methodological positioning incomplete.

Addressing these four points would strengthen both the theoretical and methodological rigor of the study, ensuring it provides a transparent, conceptually robust contribution to AI-assisted longitudinal qualitative research.

Reviewer #3: Manuscript ID: PDIG-D-26-00092

Thank you for the opportunity to review the paper entitled “Conversational, Longitudinal, Ecological Assessment (CLEA): Exploring a new AIdriven method for qualitative data collection in a behavioural health context”.

The manuscript presents a genuinely innovative study and makes a valuable methodological contribution to the field of intensive longitudinal research. However, I believe several concerns regarding methodological transparency, analytical rigor, and reflexivity require attention before the manuscript can be considered for acceptance.

--

Abstract:

1. The abstract is clear and well-written.

Introduction:

2. The introduction is well-written and informative, providing a solid overview of the relevant literature.

3. Lines 168–173 offer a brief gesture toward reflexivity and the assumptions shaping both the study and the tool development. This is a promising start, but it should not remain buried at the end of the Introduction. I encourage authors to develop this into a dedicated subsection within the Methods — addressing the composition of the research team, their theoretical assumptions, and how these shaped tool development, data collection, and analysis.

Methods:

4. The timeline of data collection remains unclear across Sections 2.2 and 2.3. While it is stated that participants could begin engaging with the tool from Week 3, the sampling frequency is never specified: Were participants prompted once per day? Every two days? How did answer deferral work in practice, and were reminders sent? These details are essential for evaluating the dataset and must be clarified.

5. It is unclear who initiated each conversational exchange. Did participants reach out to the CA, or did the CA send the first message? This has implications for understanding participant engagement and should be stated explicitly.

6. Pilot work with partners is mentioned but not adequately described. The authors should clarify the purpose and scope of the pilot, what was learned from it, and how this shaped the final design of the tool.

7. The figures displaying prompt samples are difficult to read. Beyond legibility, the authors should provide stronger rationale for the prompts used: Was prompt engineering applied? Were multiple prompt iterations tested? What criteria guided the final selection? This is an area of genuine methodological interest that deserves fuller treatment.

8. The authors may wish to include a sample of questions asked by the CA as a supplementary appendix. This would allow readers to evaluate the depth and style of the conversational data.

9. The analysis section becomes confusing as it moves back and forth between the two data types (chat logs and interviews) and the different study objectives. I encourage authors to reorganize this section using sub-headings anchored to each data type or analytical phase.

10. Inter-rater reliability statistics (e.g., Cohen's Kappa or a comparable measure) are not reported. Additionally, the rationale for dividing coding responsibility, with one author scoring chat messages and another scoring participant responses, is unclear. Could this division have introduced systematic bias into the coding procedures? Please address this directly.

Results:

11. All raters were members of the research team. The authors should reflect on how this may have influenced ratings of relevance, clarity, and utility, particularly given that participant responses were frequently brief. A brief acknowledgment of this limitation and its potential implications for data interpretation is warranted.

12. At several points the manuscript overstates the depth and richness of participant responses, even while acknowledging their brevity. The authors should review every instance in which the utility or insight of responses is characterized, and ensure the language is proportionate to what the data actually supports.

13. The Results section, like the Methods, is dense and at times difficult to follow. The segmented presentation of themes contributes to this difficulty. Clearer organization would significantly improve readability.

14. The team's reflexivity is largely absent from the Results. Given the researchers' centrality to both tool design and coding, the authors should more explicitly acknowledge how their positioning may have shaped the analysis and interpretation of findings.

15. Line 541: Please revise the phrase "to promote a sense of non-judgemental, enabling".

Discussion:

16. As in the Results, the Discussion at times overstates the usefulness and depth of participant answers. Given how little of the actual response content is presented, strong characterizations of participant insight are difficult to substantiate. The authors should moderate these claims accordingly.

17. The claim that "participants appeared to co-construct accounts with the chat-agent" (lines 582–583) is only partially supported by the findings presented. The authors should either provide stronger evidentiary grounding for this claim in the Results section, or revise the language in the Discussion to reflect the degree of support the data actually provides.

Implications of Design Choices

18. The discussion of design margins (e.g., a maximum of five questions per session) reads as a methodological description rather than a discussion of implications. This material is better placed in the Methods section, where it would help clarify the data collection procedures. The Implications section should focus on what these choices mean for the interpretation of findings and for future researchers adopting similar approaches.

Positioning CLEA Relative to Other Intensive Longitudinal Methods

19. Line 691 contains a typo: "CLEA appers Like EMA, CLEA can be" — please revise.

Overall, this is a very interesting and innovative study. Mostly, I suggest authors to further include strengthen reporting, focusing on acknowledging reflexivity/positionality; clarify the data collection timeline, sampling procedures, and initiation protocol; and reorganize the Methods and Results sections for clarity.

Reviewer #4: Abstract

The abstract currently jumps straight into the design and findings of the study, but it would benefit from including some background literature and the gap in the literature. Highlighting the theoretical and practical implications of the study would also strengthen this section.

Some parts of the author summary could be integrated into the abstract to provide a more coherent overview of what was done and to give context for the study.

Introduction

Certain sections (e.g., lines 72–74) are missing references.

Lines 74–76 read awkwardly and may contain a typo; lines 72–79 overall lack references and need revision for clarity.

Lines 82–83 mention “retrospective,” but it isn’t defined.

Lines 86–87, 103–105, 111–112, 114–115, and 160–164 are also missing references.

Careful attention to these references and smoothing out awkward sentences will make the introduction more coherent and credible.

Methods

Objectives and aims should be clearly stated in the Introduction section.

Including a detailed procedure section would enhance replicability and make it easier for readers to understand exactly how the study was conducted.

Results

Consider placing excerpts into a table under each theme. This will improve readability and make it easier to define and illustrate the themes.

Discussion

Linking the discussion of themes to past research will strengthen this section and highlight the significance of your findings.

The discussion is currently quite long and the use of subheadings could be reconsidered. Condensing this section and focusing on how findings relate to broader literature, rather than methodological comparisons, will improve clarity and impact.

6. PLOS authors have the option to publish the peer review history of their article (what does this mean?). If published, this will include your full peer review and any attached files.

**Do you want your identity to be public for this peer review?** For information about this choice, including consent withdrawal, please see our Privacy Policy.

Reviewer #1: No

Reviewer #2: No

Reviewer #3: No

Reviewer #4: No

  **Figure resubmission:**  While revising your submission, we strongly recommend that you use PLOS’s NAAS tool (https://ngplosjournals.pagemajik.ai/artanalysis) to test your figure files. NAAS can convert your figure files to the TIFF file type and meet basic requirements (such as print size, resolution), or provide you with a report on issues that do not meet our requirements and that NAAS cannot fix. 

After uploading your figures to PLOS’s NAAS tool - https://ngplosjournals.pagemajik.ai/artanalysis, NAAS will process the files provided and display the results in the "Uploaded Files" section of the page as the processing is complete. If the uploaded figures meet our requirements (or NAAS is able to fix the files to meet our requirements), the figure will be marked as "fixed" above. If NAAS is unable to fix the files, a red "failed" label will appear above. When NAAS has confirmed that the figure files meet our requirements, please download the file via the download option, and include these NAAS processed figure files when submitting your revised manuscript. **Reproducibility:** To enhance the reproducibility of your results, we recommend that authors of applicable studies deposit laboratory protocols in protocols.io, where a protocol can be assigned its own identifier (DOI) such that it can be cited independently in the future. Additionally, PLOS ONE offers an option to publish peer-reviewed clinical study protocols. Read more information on sharing protocols at https://plos.org/protocols?utm_medium=editorial-email&utm_source=authorletters&utm_campaign=protocols

---

## [Editor Report · Decision Letter 1]

28 Apr 2026

Conversational, Longitudinal, Ecological Assessment (CLEA): Exploring a new AI-driven method for qualitative data collection in a behavioural health context

PDIG-D-26-00092R1

Dear Mr Downes,

We are pleased to inform you that your manuscript 'Conversational, Longitudinal, Ecological Assessment (CLEA): Exploring a new AI-driven method for qualitative data collection in a behavioural health context' has been provisionally accepted for publication in PLOS Digital Health.

Best regards,

Sulaf Assi, PhD

Academic Editor

PLOS Digital Health

**Additional Editor Comments (if provided):**

The reviewers' comments were addressed to a satisfactory level. The manuscript now is in publishable state.